# Treatment Strategies Considering Micro-Environment and Clonal Evolution in Multiple Myeloma

**DOI:** 10.3390/cancers13020215

**Published:** 2021-01-08

**Authors:** Kazuhito Suzuki, Kaichi Nishiwaki, Shingo Yano

**Affiliations:** 1Division of Clinical Oncology/Hematology, Department of Internal Medicine, The Jikei University Kashiwa Hospital, Kashiwa-shita 163-1, Kashiwa-city, Chiba 277-8567, Japan; nishiwaki@jikei.ac.jp; 2Division of Clinical Oncology/Hematology, Department of Internal Medicine, The Jikei University School of Medicine, Nishi-shimbashi 3-25-1, Minato-ku, Tokyo 105-8461, Japan; yano@jikei.ac.jp

**Keywords:** multiple myeloma, drug resistance, bone marrow stromal cell, bone marrow niche, clonal evolution, proteasome inhibitors, immunomodulatory drugs, anti-CD38 monoclonal antibody, autologous stem cell transplantation

## Abstract

**Simple Summary:**

Multiple myeloma is an uncurable hematological malignancy, although the prognosis of myeloma patients is getting better using proteasome inhibitors (PIs), immune modulatory drugs (IMiDs), monoclonal antibodies (MoAbs), and cytotoxic agents. Drug resistance makes myeloma difficult to treat and it can be subdivided into two broad categories: de novo and acquired. De novo drug resistance is associated with the bone marrow microenvironment including bone marrow stromal cells, the vascular niche and endosteal niche. Acquired drug resistance is related to clonal evolution and non-genetic diversity. The initial treatment plays the most important role considering de novo and acquired drug resistance and should contain PIs, IMIDs, MoAbs, and autologous stem cell transplantation because these treatments improve the bone marrow microenvironment and might prevent clonal evolution via sustained deep response including minimal residual disease negativity.

**Abstract:**

Multiple myeloma is an uncurable hematological malignancy because of obtained drug resistance. Microenvironment and clonal evolution induce myeloma cells to develop de novo and acquired drug resistance, respectively. Cell adhesion-mediated drug resistance, which is induced by the interaction between myeloma and bone marrow stromal cells, and soluble factor-mediated drug resistance, which is induced by cytokines and growth factors, are two types of de novo drug resistance. The microenvironment, including conditions such as hypoxia, vascular and endosteal niches, contributes toward de novo drug resistance. Clonal evolution was associated with acquired drug resistance and classified as branching, linear, and neutral evolutions. The branching evolution is dependent on the microenvironment and escape of immunological surveillance while the linear and neutral evolution is independent of the microenvironment and associated with aggressive recurrence and poor prognosis. Proteasome inhibitors (PIs), immunomodulatory drugs (IMiDs), monoclonal antibody agents (MoAbs), and autologous stem cell transplantation (ASCT) have improved prognosis of myeloma via improvement of the microenvironment. The initial treatment plays the most important role considering de novo and acquired drug resistance and should contain PIs, IMIDs, MoAb and ASCT. This review summarizes the role of anti-myeloma agents for microenvironment and clonal evolution and treatment strategies to overcome drug resistance.

## 1. Introduction

Multiple myeloma (MM) is a type of hematological malignancy that is often uncurable because it comprises a heterogeneous group of plasma cell neoplasms which vary in terms of their morphology, phenotype, molecular biology, and clinical behavior [1]. Drug resistance is one such clinical behavior that makes MM difficult to treat, and it can be subdivided into two broad categories: de novo and acquired. De novo resistance includes environment-mediated drug resistance, which is categorized as either cell adhesion-mediated drug resistance (CAM-DR) or soluble factor-mediated drug resistance (SFM-DR). These types of drug resistance are induced rapidly via the events of a signaling pathway concerning tumor microenvironments [2,3,4]. Interactions between myeloma cells and bone marrow stromal cells (BMSC) or extracellular matrix proteins promote the growth, survival, migration, and drug resistance of tumor cells. The adhesion of myeloma cells to BMSC induces the secretion of several cytokines and growth factors, such as interleukin (IL)-6, and insulin-like growth factor-1 (IGF-1). These cytokines and growth factors are produced and secreted by myeloma cells in the bone marrow (BM) microenvironment and are regulated by autocrine and paracrine loops [5]. MM-associated microenvironments are categorized into “vascular niche” and “endosteal niche” [6]. The vascular niche is constituted via endothelial activation by mainly vascular endothelial growth factor (VEGF)/VEGF receptor (VEGFR) interaction, and contributes to the migration, proliferation, and survival of MM cells. Hypoxia contributes to the development of a vascular niche and the maintenance of stemness of MM cells. The endosteal niche, which is constituted of mainly osteoclasts and osteoblasts, contributes to the survival and dormancy of MM cells [7,8].

Acquired drug resistance develops over time as a result of sequential genetic changes that ultimately culminate in complex therapy-resistant phenotypes [2]. Clonal evolution is a form of genetic change concerning acquired drug resistance. Clonal evolution is divided into [1] branching clonal evolution, which is dependent on the microenvironment, and [2] linear and [3] neutral clonal evolution, which are both independent from the microenvironment [9]. Maintenance of a deep response, such as minimal residual disease (MRD) negativity, is essential for the prevention of aggressive clonal evolution [10,11]. Non-genetic diversity is considered to be an acquired drug resistance, such as an immunologic phenotypic change in the target molecules of monoclonal antibodies agents (MoABs) [12,13].

Finally, four classes of anti-myeloma agents are available—proteasome inhibitors (PIs), immune modulatory drugs (IMiDs), MoABs, and cytotoxic agents—but MM is an uncurable hematological malignancy due to de novo and acquired drug resistance, although the prognosis of myeloma patients is getting better using these anti-myeloma agents. Overcoming de novo and acquired drug resistance plays an important role in the search for a clinical cure in myeloma patients. High-dose chemotherapy followed by autologous stem cell transplantation (ASCT) has been a standard of care for fit, newly diagnosed MM (NDMM) patients, since it not only relieves hematopoiesis but also affects the microenvironment. Initial treatment is extremely crucial because the microenvironment is relatively better and the frequency of clonal evolution is lower in NDMM than relapsed or refractory MM (RRMM), and should include treatment using multi-class drugs including ASCT for fit NDMM patients to help overcome de novo drug resistance and prevent acquired drug resistance. Here, we discuss treatment strategies for MM concerning de novo and acquired drug resistance. The relationship between MM cells and BMSC, the vascular niche, and endosteal niche and treatment for microenvironment are summarized in Figure 1. Clinical trials of new agents for target concerning bone marrow microenvironments in MM are shown in Table 1.

## 2. Interaction with Bone Marrow Stromal Cell

### 2.1. Cell Adhesion-Mediated Drug Resistance and Soluble Factor-Mediated Drug Resistance

Cell adhesion-mediated drug resistance (CAM-DR) is induced by the adhesion of tumor cell integrins to stromal fibroblasts or to components of the extracellular matrix, such as fibronectin, laminin, and collagen [2]. The adhesion molecules, such as very late angine-4 (VLA-4) plays an important role for CAM-DR [26]. VLA-4 is made of a heterodimer of CD49d/CD29 on MM cells. Interaction between VLA-4 and vascular cell adhesion molecule-1 (VCAM-1) binds between MM cells and BMSC, contributing to the survival of MM cells via activation of phosphoinositide 3-kinase (PI3K)/(protein kinase B) AKT pathway and CAM-DR [14]. The epigenetic mechanism is associated with CAM-DR as well. The phosphorylation-mediated enhancer of zeste homolog 2 (EZH2) inactivation and subsequent decreases in H3K27me3 levels are related to CAM-DR in MM cells [15]. Thus, EZH2 is a target of treatment for the apoptosis of myeloma cells and release of CAM-DR [27,28]. Inhibition of EZH is known to inactivate CAM-DR in vitro [27]. CAM-DR is induced in fibroblasts derived from MM patients, but not in healthy individuals [29,30]. CAM-DR may be strong in minimal residual myeloma cells compared to newly diagnosed myeloma cells because of the rich presence of adhesion molecules on the minimal residual myeloma cells [31].

Soluble factor-mediated drug resistance (SFM-DR) is induced by cytokines, chemokines and growth factors secreted by fibroblast-like tumor stromal cells [2]. BMSC-derived soluble factors, such as IL-6, IGF-1, and stromal cell-derived factor-1 (SDF-1), induce the development, proliferation, and survival of MM cells via the adhesion of the microenvironment by signal transduction such as nuclear factor-kappa B (NK-kB), Janus kinase (JAK)/signal transduces and activator of transcription 3 (STAT3), RAS/RAF, and PI3K/AKT pathway [2,32]. IL-6 and IGF-1 cooperate to enhance growth of MM cells [33]. IL-6 can trigger the binding of the membrane IL-6 receptor (IL-6R) to the IGF-1 receptor (IGF-1R) and can induce IGF-1R phosphorylation independently of the addition of IGF-1 [33].

IL-6 is an inflammatory cytokine with the ability to induce tumor growth, metastasis, and resistance to chemotherapy in a variety of tumor cells [34]. IL-6 acts as an antiapoptotic mediator in myeloma cells, whereas in normal cells, its functions seem to mainly involve cell differentiation and development [35]. The level of serum IL-6 is significantly higher in NDMM patients than RRMM [36]. A high expression of IL-6 receptors can indicate a shorter survival time in the NDMM patients treated with the Total Therapy 2 protocol [37]. The ratio of IL-6 present was higher in the patients with high-risk cytogenetic abnormality (HRCA), such as t(4;14), del13q, and 1q21 gain, than those without HRCA. In contrast, the expression of IL-6 receptors was lower in the patients with t(11;14) than those without t(11;14) [38].

IGF-I is single chain polypeptide made of 70 amino acids that functions as the primary ligand for IGF-IR [39]. IGF-1, which are secreted from BMSC and autocrined by MM cells, binds with IGF-1R, contributing to the proliferation of myeloma cells and CAM-DR [27,40]. IGF-1 promotes growth and proliferation of myeloma cells via activation of the IGF-1 receptor in addition to IL-6 [41,42]. However, the serum IGF-1 levels were similar in NDMM patients and healthy donors despite high serum IGF-1 levels being associated with shorter overall survival (OS) before proteasome inhibitors (PIs), immune modulatory drugs (IMiDs) were available [43]. IGF-1 induces protein synthesis, enhances endoplasmic reticulum (ER) stress in myeloma cells, and enhances the activity of proteasome inhibitors (PIs) [44]. The presence of IGF-1 receptors has predicted short survival times in the NDMM patients treated with the Total Therapy 2 protocol [37]. The ratio of IGF-1R present was higher in the patients with HRCA, such as t(4;14), del13q, del17p, and 1q21 gain, than those without HRCA. The ratio of IGF-1R present was similar between the patients with and without t(11;14) [36]. IGF-1 increases the expression of hypoxia-inducible factor-1 (HIF-1) via the activation of the AKT and mitogen-activated protein kinase (MAPK) pathways [45], and results in the secretion of VEGF [34]. IGF-1 regulates anti-apoptotic protein B cell lymphoma 2 (BCL2) in myeloma [38]. Thus, interaction between IGF-1 and IGF-1R is associated with angiogenesis and survival of myeloma cells, and IGF-1R is considered a therapeutic target.

### 2.2. Treatment for Interaction with CAM-DR and SFM-DR

CAM-DR and SFM-DR are therapeutic targets to overcome drug resistance and to suppress the proliferation and survival of MM cells via the inhibition of signaling pathways.

Bortezomib (BOR) overcomes CAM-DR by inhibiting VLA-4 [26], and shows a synergistic effect with various drugs in co-culturing human myeloma cell lines with BMSC [46]. Proteasome inhibition suppresses cytokine-mediated survival and proliferation advantages by inhibiting the NF-kB pathway in BMSCs [47]. Interactions between MM cells and BMSCs induce resistance for immunomodulatory drugs (IMiDs) by decreasing the expression of IKAROS family zinc finger 1 (IKZF1), which is a biomarker for sensitivity to IMiDs [48,49]. BOR enhances the activity of IMiDs by releasing CAM-DR and elevating IKZF1 expression.

Cluster of differentiation (CD)38 expression on the surface of myeloma cells is down-regulated when co-cultured with BMSCs due to the CD38 on myeloma cells binding to CD31 on BMSCs [50]. BOR inhibits CD38-CD31-mediated adhesion of myeloma cells to BMSCs and increases CD38 expression on myeloma cells by decreasing CD31 expression on BMSC [51]. Thus, BOR might enhance the anti-myeloma effect of anti-CD38 MoAb such as daratumumab (DARA) via the inhibition of adhesion between myeloma cells and BMSCs. DARA releases CAM-DR by inhibiting adhesion to BMSCs by internalizing CD38 in myeloma cells [52].

## 3. Myeloma Cells in Hypoxia and Vascular Niche

### 3.1. Vascular Niche in Myeloma

MM-associated niches are categorized into the “vascular niche” and “endosteal niche”, which are associated with the vasculature and osteopoiesis, respectively [53]. The endothelium plays an important role in formation of the vascular niche and maintenance of hematopoiesis [54,55]. The vascular niche enhances the metastatic outgrowth not only by delivery of oxygen, nutrients, and several growth factors but also via interaction with tumor cells [54]. In the early stages of bone metastasis, the vascular microenvironment could fail in the homing and colonization of circulating myeloma cells in the BM. In the later stage, the vasculature reduces metastatic growth, but does not induce the disappearance of BM metastases. BM blood has a lower oxygen partial pressure than venous blood, and the vascular niche may even have a much lower oxygen content [56,57]. Hypoxia plays an important role in myeloma pathogenesis [58].

The interaction between VEGFs, which is composed of six members, and VEGFRs promotes endothelium regeneration [59]. In MM, interaction VEGF-A/VEGFR-2 induces migration, proliferation, and survival of MM cells via autocrine and paracrine of VEGF [60,61,62,63]. VEGF promotes micro-angiogenesis of BM via activation of PI3K/AKT pathway, and is associated with poor prognosis in myeloma [64]. VEGF is mainly regulated by HIF, which is transcriptional factor activating in hypoxia [65,66,67].

#### Hypoxia in Myeloma

Hypoxic stress is considered to give stem cell-like properties to cancer cells [68]. Mean oxygen saturation in bone marrow is 87.5% ± 1.1% when that in peripheral blood is 99% in healthy volunteers [69]. A specific marker for cancer stem cells is still unclear, but it may be present in the CD138-negative fraction in myeloma cells. Hypoxic stress increases stem cell markers, such as octamer-binding transcription factor 4 (Oct-4), NANOG and (sex determining region Y (SRY)-box 2) SOX2, in CD138-negative myeloma cells [68]. Myeloma stem cells are considered to be present in hypoxic BM, and to have drug resistance [70].

Hypoxia prevents myeloma cells from maturing and might reduce the expression of signaling lymphocytic activation molecule family member 7 (SLAMF7) and CD38, which are the target of elotuzumab (ELO) and DARA or isatuximab (ISA), depending on non-genetic diversity, which is described in detail in Section 5.6, “Nongenetic clonal diversity” [71,72,73]. Myeloma cells without CD138 expression show resistance to BOR [70]. These changes in myeloma cells that prevent maturation might be induced by the suppression of interferon regulatory factor 4 (IRF4) by hypoxic stress [68,74].

Hypoxia decreases E-cadherin expression, making MM cells move from adhesion of BMSC into peripheral blood (PB) [75]. In addition, hypoxia increases C-X-C chemokine receptor type 4 (CXCR4) expression on MM cell and decreases SDF-1 from stromal cells, inducing MM cells to migrate to new BM niches [75,76,77]. Finally, MM cells lose the dependence on the microenvironment, engraft extramedullary, and form extramedullary disease (EMD) [78,79].

HIF-1alfa is the master regulator of oxygen homeostasis [46,59,67,70,80]. The HIF-1alfa protein is degraded by the ubiquitin-proteasome pathway under hypoxic condition [65]. HIF-1a is downstream of the PI3K/AKT/mammalian target of rapamycin (mTOR) pathway and is associated with the progression of myeloma cells through the transcription of several important genes concerning myeloma pathogenesis, such as VEGF and IGF-1 [81]. HIF-1alfa is known to have formed positive feedback with MYC [82] and elevated expression of IRF4 [83]. A high level of HIF-1alfa derived from endothelial cells is a poor prognostic factor in MM [84]. HIF-1alfa increases suppressive TIL (tumor infiltrating lymphocytes) such as Treg [85]. Thus, HIF-1alfa is a target molecule for myeloma treatment.

### 3.2. Treatment for Vascular Niche and Hypoxia

Treatment for hypoxia and vascular niche plays an important role in preventing the occurrence of EMD and maintaining the immature MM cells.

BOR suppresses transcription of HIF-1alfa [86] and indirectly inhibits MM angiogenesis by decreasing the level of VEGF [87]. IMiDs, especially thalidomide (THAL), inhibit angiogenesis by suppressing transcription of the VEGF gene [88,89,90]. Lenalidomide (LEN) inhibits VEGF-induced PI3K/Akt pathway signaling and HIF-1alfa expression on endothelial cells [91].

Hypoxia plays an important role in EMD formation [75] and is shown as an uptake using positron emission tomography-computed tomography (PET-CT) [92]. THAL plus dexamethasone (DEX) has not been proven effective for EMD in RRMM [93]. VTD (BOR + THAL + DEX) and VRd (BOR + LEN + DEX) followed by ASCT were evaluated in the CASIOPEIA and IFM2009 trials, although there has not yet been a clinical trial that has made a direct comparison between VTD and VRd [94,95]. Complete remission (CR) ratio of VTD was lower than those of VRd (38.5% vs. 59%), while the imaging CR ratio of VTD was similar to that of VRd using PET-CT (63.9% vs. 62%) [96,97]. Thus, VTD is considered to be effective for myeloma in hypoxia due to the release of EMD by THAL although hypoxia induces resistance for BOR [98].

Finally, VDT-PACE (BOR, THAL, DEX, cisplatin, doxorubicin, cyclophosphamide, plus etoposide) might be effective for patients with HRCA and extramedullary plasmacytoma retrospectively [99]. In this regimen, THAL enhances the sensitivity of BOR and cytotoxic agents by improving hypoxia, and BOR, in turn, increases the sensitivity of doxorubicin and etoposide, whose activities are dependent on the cell cycle, through the release of CAM-DR.

## 4. Endosteal Niche Contributes Survival of Myeloma Cells

### 4.1. Endosteal Niche is Constituted by Osteoclasts and Osteoblasts

The endosteal niche is constituted by osteoclasts and osteoblasts, and is a reasonable environment to enable the survival of myeloma cells [100]. Osteoclasts contribute to the survival and proliferation of myeloma cells by releasing the receptor activator of nuclear factor kappa-B ligand (RANKL), which is secreted by myeloma cells [101], osteoclasts [102], and BMSCs [103], and the abnormality of RANΚL-osteoprotegerin (OPG) pathway [104,105,106]. An elevated level of RANKL is associated with increasing bone lytic lesions and short survival of myeloma patients [107]. Osteoclasts suppress the immune effect of cytotoxic T cells (CTL) [108]. The acidic condition induced by osteoclasts activate the PI3K/AKT pathway in MM cells [109].

Mature osteoblasts suppress the survival and proliferation of MM cells. Soluble inhibitors of the canonical Wnt pathway such as Dickkopf-1 (DKK-1) produced by MM cells suppress the differentiation of osteoblasts [110]. DKK-1 regulates osteoclastgenesis [111] and osteoblastgenesis directly [112]. The serum level of DKK-1 is high in MM patients with bone lytic lesions, and decreased when treatment including ASCT and PIs was effective [113,114,115]. Finally, osteoblasts contribute to maintaining the dormancy of MM cells, inducing resistance to melphalan and BOR [7,116].

The endosteal niche parallels the vascular niche in regulating hematopoiesis and tumor survival or dormancy during the metastatic cascade [7,8,116]. Residual MM cells, after ASCT, escape from the vascular niche in order to protect themselves from high concentrations of melphalan [117]. Thus, the endosteal niche is one of important therapeutic targets to eradicate MM cells.

### 4.2. Treatment for Endosteal Niche

Bone remodeling by treatment for the endosteal niche is essential for MM cells. BOR enhances osteogenesis via the inhibition of several factors concerning osteolytic disease, such as RANKL and DKK-1 and activation of osteoblasts by stabilizing β-catenin, which is essential for the differentiation of osteoblasts [115]. DARA suppresses osteolysis by osteoclasts [118]. ASCT contributes to bone remodeling as well [119].

RANKL and DKK-1 are therapeutic targets. Denosumab, which is an anti-RANKL MoAb, prevents skeletal-related events and prolongs PFS in the MM patients with bone lytic lesion compared with zoledronic acid [23]. BHQ880, which is an anti-DKK-1 MoAb, suppresses the progression of myeloma cells in vitro and in vivo [120].

## 5. Clonal Evolution

### 5.1. Two Types of Driver Mutation

Myeloma stem cells are considered to be generated due to the acquisition of 14q chromosomal translocation or hyperdiploidy when class switch and somatic mutation occur in post-germinal center cells. The 14q chromosomal translocations are commonly associated with aberrant gene expression of the cyclin D family, which confers a growth advantage to dormant plasma cells or post germinal center (post-GC) B-lymphocytes. Approximately 10% of cases overlap 14q translocation and hyperdiploidy [121].

Multiple ancestor clones are first produced from myeloma stem cells by the ‘big bang’, which means tumors grow accompanied by not only the main clone but also by numerous subclones without stringent clonal selection [122]. In each clone, genetic abnormalities are added to the reservoir myeloma clones by selective pressures by treatment, causing branching clonal evolution [123]. The type of driver mutation affects the clonal evolution in MM. MM cells carrying HRCA, such as t(4;14), acquire more copy number abnormalities than those with standard-risk abnormalities, such as t(11;14), during the course of disease progression [124,125].

### 5.2. Three Types of Clonal Evolution

Clonal evolution is a process of clonal expansion and clonal selection through genetic change within the adaptive condition. Clonal evolution can be divided into branching, linear, and neutral clonal evolutions [9].

Branching clonal evolution is defined as different clones having no superiority or inferiority, as one clone does not destroy the other clone, and each clone progresses independently when abnormalities of driver genes occur in different clones at the same time. In the branching evolution, driver genes associated with tumor development show heterogeneity in the local site of myeloma. The incidence of branching clonal evolution is associated with the fitness of different clones to BM microenvironments including immune systems and treatment pressure [123]. For example, clones undergo enhancement of IL-6 receptor and programmed death 1-ligand 1 (PD-L1) genes, resulting in growth and proliferation of myeloma cells and escape from immune surveyance [123].

The linear clonal evolution is defined as clones with established mutations growing and coexisting without any apparent selective pressure. The linear clonal evolution is called the “big bang model” clonal evolution, and related to genomic instability [122,123]. In RRMM, especially plasma cell leukemia (PCL), with linear clonal evolution, genes affecting tumor phenotypes such as cell adhesion or tumor suppressor genes is pointed out [126,127,128,129].

Neutral evolution is a kind of clonal evolution as stated above. The majority of mutations are not beneficial but are evolutionarily neutral, and do not contribute to variation at the molecular level due to their rapid elimination by natural selection. Therefore, mutant alleles can be accidentally modified as a result of “survival of the luckiest” rather than by a selective advantage [123,130]. The frequency of neutral evolution was higher in the patients with IgH translocation than those with hyperdiploidy [131].

### 5.3. Genetic Change during Treatment in PCL and EMD

The terminal stage of myeloma is independent from BMSC, which results in the formation of extramedullary lesions and leukemic change. EMD including PCL is a less frequent manifestation of myeloma, which is defined as the infiltration organ or circulation of PB independent from BMSC [78]. EMD is more frequent in patients with RRMM than those with NDMM [78], and is considered to be more frequent in the terminal stage of myeloma and due to genetic changes [79]. The frequency of IgH loci translocations is similar between the patient with NDMM and EMD, while additional cytogenetic abnormalities, such as del17p and 1q21 gain, and gene mutation, such as TP53 and RAS, increased in the patients with EMD compared with those with NDMM [79]. The incidence of EMD as a recurrence was higher in the patients with HRCA than those without HRCA [132]. There was no consensus regarding the association between the incidence of EMD and the selection of initial treatment [79].

In the terminal stage of myeloma, homozygous deletion of the genes associated with inhibition for the NF-kB pathways is detected (10–15% in total), including baculoviral IAP repeat containing protein 2/3 (BIRC2/3) on chromosome 11 (~7%), TNF receptor-associated factor 3 (TRAF3) on chromosome 14 (~3%) and cylindromatosis (CYLD) on chromosome 16 (~3%) [133].

The loss of 17p is observed in approximately 10% of patients with NDMM and increases in frequency along with disease progression in the 50–70% of MM cases with EMD and PCL. TP53, a tumor suppressor gene, is located at 17p13 and is monoallelic in the majority of MM patients. p53 haploinsufficiency induces failure of DNA repair, which is related to acquired point mutations in the other allele of TP53 during disease progression. TP53 mutations generally is observed in patients with del17p in MM [134]. TP53 mutations are detected in around 33% of patients with del17p in NDMM, and about 50% of patients with del17p in RRMM, respectively [135,136]. Finally, the frequency of TP53 mutations is 5–8% in NDMM [136,137] and up to 25% in PCL [132,138]. In addition, mutant p53 usually has oncogenic functions, such as the upregulation of c-Myc and genes encoding proteasome subunits [139], which can induce anti-cancer drug resistance [140,141,142]. As far has reports go, there has been no incidence of a treatment that could truly overcome the poor prognosis of del17p, although several new anti-myeloma agents improved the clinical outcome in the patients with del17p compared with old agents, and it remains an unmet need for the treatment of myeloma [143].

### 5.4. HRCA is Associated with Clonal Evolution

HRCA was identified from different BM sites at the same time in 25% of patients diagnosed with non HRCA [144]. High uptake using fluorodeoxyglucose (FDG)-PET/CT was associated with HRCA [145]. Genomic instability is associated with increasing driver mutations, loss of heterozygosity (LOH), and APOBEC signatures and predicts poor prognosis [144,146,147]. In particular, t(14;16) and t(14;20) is associated with the APOBEC signature. LOH is associated with the APOBEC signature and P53 deletion [146,148].

DIS3 mutations are associated with t(4;14), and a large number of DIS3 mutations in minor clones can predict a poor prognosis [135]. It is considered that this is because the clone with the DIS3 gene mutation was selected by the clone tide, and the growth advantage was obtained [148]. The incidence of neutral evolution in the MM cases with 14q translocations, especially t(4;14) and t(4;16), was significantly higher than that of MM cases driven by hyperdiploidy as well [131] because t(4;14) and t(14;16) accelerate clonal heterogeneity and lead to a clonal dominance of selected clones [124,149,150]. Neutral evolution was associated with TP53 gene mutation and 1q21 gain, and predicts poor prognosis in patients treated with IMiDs [123,131]. THAL induced clone selection more frequently than BOR [150]. ASCT might overcome the negative impact of neutral clonal evolution [131]. Thus, PIs-containing chemotherapy and ASCT might be essential for patients with HRCA considering the incidence of neutral clonal evolution.

### 5.5. Relation between Deep Response and Clonal Evolution

In MRC Myeloma XI, the incidence of gene mutation in the patients with PR was significantly higher than those with very good partial response (VGPR) and CR. The incidences of linear evolution and branching evolution were almost similar, regardless of the therapeutic effect. OS after recurrence was not associated with the type of clonal evolution [151]. According to analysis of clone selection using fluorescence in situ hybridization (FISH) from China, the recurrences accompanied with clonal evolution are observed frequently in HRCA (especially del17p, 1q21gain) [150]. Clone selection was pointed out infrequently in the patients with relapsed myeloma from MRD-negativity and CR. These results may be opposite to those of MRC Myeloma XI concerning the relation between the incidence of clonal evolution and the depth of response.

These two reports mention that the incidence of clonal evolution depends on a deep response, while there is no evidence about association between clonal evolution and the period during MRD negativity. MRD-negativity was identified as a surrogate marker for PFS prolongation [10,11]. Several reports showed that a long interval of MRD-negativity predicted long PFS, while aggressive relapse, such as the presence of EMD, might occur after MRD-negativity to clonal evolution [152]. Considering the incidence of developing clonal evolution, we believe that it is necessary to continue treatment to prevent recurrence even after achieving MRD-negativity especially in the patients with HRCA.

### 5.6. Nongenetic Clonal Diversity

Clonal evolution induces clonal diversity and is a cause of acquired drug resistance, while conversion of phenotype is a cause of drug resistance such as nongenetic clonal change in myeloma. Chaidos et al. demonstrated that the ratios of CD19^−^CD138^−^ pre-PC (plasma cell) myeloma cell, CD19^−^CD138^low^ myeloma cell, and CD19^-^CD138^high^ myeloma cell were changed via therapeutic selection and pre-PC myeloma showed drug resistance [153]. Paiva et al. demonstrated that CD19^−^CD81^+^ myeloma cell differentiated into CD19^−^CD81^−^ mature myeloma cell in 5% of patients, while CD19^−^CD81^−^ mature myeloma cell changed into CD19^−^CD81^+^ myeloma cell in 20% of patients via therapeutic selection in the myeloma patients treated with bortezomib, melphalan, and prednisone (VMP) [12]. The CD38 expression, which was associated with a clinical response of DARA, was down after the start of DARA, and elevated 6 months after the conclusion of DARA [13]. Thus, the phenotype should be analyzed before a change in chemotherapy, because the nongenetic clonal change often occurs.

IMiDs might be effective dormant myeloma cells under the BM niche because dormant MM cells, which adhere to the BM niche, highly express IKZF1 [154]. LEN is active for immature myeloma cells and side population [155,156,157]. In contrast, PIs are effective mature myeloma cells which have ER stress [68,71]. Thus, IMiDs and PIs work for myeloma cells in different differentiation stages.

CD38 and CD319, which is identified with SLAMF-7, are expressed independently of maturation of myeloma cells [153]. Thus, anti CD38 and SLAMF-7 MoAbs can be active for myeloma cells in various differentiation stages.

## 6. Overall Treatment Strategy Considering Microenvironment and Clonal Evolution

### 6.1. Concept of Treatment Concerning Microenvironment and Clonal Evolution

MM is considered to be uncurable because MM cells are protected by the BM microenvironment, including CAM-DR, SFM-DR, vascular and endosteal niches, and clonal evolution leads to refractoriness for chemotherapy.

Initial treatment should be given the most importance in treatments approaching de novo and acquired drug resistance. First, the majority of molecules concerning SFM-DR increase, dependent on disease progression from MGUS, NDMM, and RRMM to EMM. For example, IL-6 increases in RRMM, and suppresses CD38 expression on MM cells, inducing resistance to DARA. Second, the maintenance of a deep response should be essential to prevent clonal evolution. Several clinical trials have demonstrated that a short interval of MRD-negativity is not adequate for long survival [10,11]. Meanwhile, how long an interval of MRD-negativity is enough to achieve a clinical cure has been open to discussion until now. In addition, the incidence of EMD was higher in the patients with RRMM than NDMM after MRD-negativity was failed [152]. Third, immune reconstitution can predict a good prognosis in the patients with MM independently from MRD and HRCA [157,158,159,160,161,162,163,164]. Finally, the microenvironment, such as CAM-DR, hypoxia, and the endosteal niche, contributes to the survival of MRD and MM stem cells. Treatment of the microenvironment is necessary to kill all myeloma cells. Associations between the activity of treatments, including PIs, IMiDs, anti CD38 MoAb, and ASCT, cytogenetic risk and maturation of MM cells are summarized in Figure 2 [123,165,166]. Treatment strategies for fit and unfit patients with NDMM, and RRMM, terminal phase MM are shown in Figure 3.

PIs release CAM-DR via the inhibition of VLA-4, induce ER stress for M protein rich mature myeloma cells, improve hypoxic condition, and promote bone remodeling as above [68]. High expression of XBP1, which is associated with the differentiation of plasma cells and independent from cytogenetic risk in MM, predicted good clinical outcome in MM patients treated with BOR via ER stress [167,168,169]. Thus, PIs might be active for MM independently from cytogenetic risk. In addition, PIs enhance ADCC activity by suppression of HLA class 1 expression on MM cells [170].

IMiDs enhance both adaptive and innate immune system via co-stimulation of T cells and enhancement of NK and NKT cells in vitro [171]. IMiDs suppress Treg function by suppression of Foxp3 gene expression in Treg [172]. Finally, IMiDs suppress PD-1 expression on T and NK cells [173], and LEN suppresses PD-L1 expression in MM cells [174]. Thus, LEN and pomalidomide (POM) are good partners of some MoAbs because they enhance antibody-dependent cellular cytotoxicity (ADCC). IMiDs contributes to immune reconstitution compared with PIs [164]. High IKZF1 expression, which was higher in non-HRCA than HRCA, predicted a good treatment response of IMiDs [175]. In addition, IMiDs tended to be active for immature MM cells from the results of genetic analyses and clinical trials compared with PIs [68].

Anti CD38 MoAbs have several anti-myeloma and microenvironment activities, such as complement-dependent cytotoxicity (CDC), ADCC, direct myeloma killing, enhancement of the immune system, inhibition of CAM-DR, and enhancement of bone remodeling as above. CD38 expression is independent of the maturation of MM cells’ cytogenetic abnormality [153,176], but is dependent on serum IL-6 level, which increases in disease progression [177]. Anti-CD38 MoAbs contribute to the enhancement of the immune system [178]. The activity of anti CD38 MoAbs in NDMM should be higher than RRMM because the immune system in NDMM is better than those in RRMM [179]. The efficacy of DARA for HRCA is still controversial. DRd (DARA + LEN + DEX) contributed to the longer progression-free survival (PFS) compared with Rd in a POLLUX trial, which is a phase 3 trial about DRd versus Rd for RRMM; the hazard ratio for PFS of DRd compared with Rd decreased as the follow-up interval increased [180,181,182]. DRd can be active for HRCA, excluding the patients at ultra-high risk, whose diseases progress in the early stages after DRd started. It was suggested that DRd continuous treatment might be effective for HRCA patients. Thus, anti-CD38 MoAbs contribute to anti-myeloma activity in early phase MM cells.

### 6.2. Autologous Stem Cell Transplantation for Microenvironment and Clonal Evolution

High-dose melphalan followed by ASCT is still a standard of care. High dose melphalan kills MM cells nonspecifically via deoxyribonucleic acid (DNA) damage and may attack the subclonal structure, potentially reducing the impact of a neutral or non-neutral evolutional tumor history [123,131]. In addition, the autograft plays an important role for the therapeutic effect on not only the recovery of myelosuppression but also the improvement of microenvironment in the patients received ASCT. Mesenchymal stem cells (MSC), which are non-hematopoietic undifferentiated cells, are able to differentiate into not only components of BMSC such as hematopoietic tissues, but also various mesodermal-type tissues such as osteoblasts, chondrocytes, and myocytes [183]. BMSC are a major component of the hematopoietic microenvironment, and contribute to maintaining long-term hematopoietic maintenance. In addition, osteoblasts work as the actual state of endosteal niches. Thus, MSC play an important role as one of the major members responsible for the hematopoietic mechanism. MSC are difficult to collect from the PB of healthy individuals, while they are also mobilized in PB when mobilized with granulocyte colony stimulating factor (G-CSF) [9].

In FORTE trial, PFS in the patients treated with four cycles of KRd (carfilzomib (CFZ) + LEN + DEX) followed by ASCT followed by four cycles of KRd (KRd-ASCT group) was significantly longer than those with 12 cycles KRd (KRd12), although the MRD negativity ratio was similar using multicolor flow cytometry (cut-off = 10^−5^) [184,185]. In addition, in a long follow-up, the MRD ratios were similar between the KRd12 and KCd (CFZ + cyclophosphamide + DEX)-ASCT groups. The reason why KRd12 was inferior compared with KRd-ASCT and similar with KCd-ASCT is a deeper response (MRD negativity rate using next generation sequencing, 23% vs. 34%; cut-off = 10^−6^) and immune reconstitution in the patients that received ASCT. Thus, up-front ASCT is a standard of care from the point of view concerning the results of clinical trials but also the improvement of the microenvironment.

### 6.3. Clinical Significance of Immune Reconstitution

Immune reconstitution is important for the patients with HRCA instead of MRD status. Continuous IMiD treatments induce immune reconstitution, and play an important role for the treatment for myeloma patients with HRCA. However, survival time in the HRCA patients treated with IMiDs was short in the majority of clinical trials. Continuous therapy with IMiDs combined with PIs or monoclonal antibodies might be essential for myeloma patients with HRCA. In addition, ASCT contributes to a deeper response and the improvement of the microenvironment instead of neutral evolution and HRCA [131]. In the patients that received ASCT, normal plasma cells increase in patients with MRD-negativity compared with those with MRD-positivity, leading to immune reconstitution [105]. Because the benign BM niche provides support to the survival of normal plasma cells, regulates the number of normal plasma cells [186], and promotes immune reconstitution, it is suggested that the benign BM niche might increase via the eradication of MM cells in niche by intensive chemotherapy, increasing the niche available for normal plasma cells [117,187]. Therefore, multi-drugs approaches, PIs, IMiDs, DARA, and ASCT, might be essential for fit NDMM patients from the point of view of the microenvironment. In fact, DARA + VRd followed by ASCT was tolerable and effective for NDMM in GRIFFIN trial [188].

### 6.4. Treatment for Unfit and Frail NDMM, RRMM, and EMDs

In the case of unfit or frail NDMM patients who are ineligible for intensive treatment including ASCT, DARA-based treatment is a good option because CD38 expresses on MM cells independently from cytogenetics [189], and a relatively better immune system contributes the activity of anti-CD38 MoAbs [190,191,192,193]. Partners of DARA, such as PIs and IMiDs, are selected considering cytogenetic abnormality, clinical symptom, and comorbidity. Differentiation of MM cells and the cancer panel should contribute to classifying which MM cells are suitable for treatment with PIs or IMiDs [68,194,195]. For frail patients, doublet regimens, such as VD (BOR + DEX) and Rd might be selected considering the general condition, comorbidity, and social background rather than cytogenetic risk. There is no evidence of the efficacy and tolerability of anti-CD38 MoAbs plus PIs or IMiDs as initial chemotherapy for frail patients up to now, although PFS in the patients treated with DARA + VMP and DARA + Rd were significantly longer than those in those treated with VMP and Rd among the elderly patients according to ALCYONE and MAIA trials, respectively [196]. If frailty improved via the efficacy of chemotherapy, anti-CD38 MoAb can be added. Initial treatment should be continued until PD or unacceptable adverse events, because the possibility of receiving the next chemotherapy is low [197].

In the RRMM patients after the multi-drug approach, the reason why treatment failed should be evaluated. Fluorescence in situ hybridization (FISH) for del17p, 1q21 gain, and MYC containing a cytogenetic abnormality should be repeated for each relapse because these cytogenetic abnormalities can be added. The phenotype of MM cells using flow cytometry should be repeated to analyze non-genetic changes to MM cells. Thereafter, the suitable treatment for cytogenetic abnormality and phenotype should be selected.

In the terminal phase of MM, EMD is often detected. The treatment for EMD is challenging. Cytotoxic agents containing chemotherapy including ASCT might be effective for EMD [79,198,199]. VTD-PACE is a good option for EMD considering CAM-DR and hypoxia, but might be intolerable for the majority of patients with a terminal phase of MM [200].

### 6.5. Future Direction

Several new drugs are developed to overcome de novo drug resistance, such as proteasome inhibitors and EZH2 inhibitors to overcome CAM-DR, and anti-IL-6 monoclonal antibodies and IGF-1R inhibitors to overcome SFM-DR, in a preclinical study, although several of those have not shown efficacy in clinical trials and thus are not available in clinical practice. In addition, treatments to prevent and overcome acquired drug resistance are already one of unmet medical needs as well. In future, treatment strategies to prevent clonal evolution and overcome acquired drug resistance including the terminal phase of myeloma will be necessary. Treatment with new modes of action, such as chimeric antigen receptor-T cell therapy (CAR-T) [201], antibody-drug conjugate (ADC) [202], bispecific T-cell engager (BiTE) for B cell mature antigen (BCMA) [203], exportin-1 (XPO-1) inhibitor [204] and melflufen [205], may be active for patients with a terminal phase of MM.

## 7. Conclusions

This review summarizes myeloma pathogenesis and drug resistance from the perspectives of the adhesion of BMSCs, hypoxia, and vascular and endosteal niches, and describes treatment strategies for each microenvironment in myeloma. The research and development of specific inhibitors for the BM microenvironment are progressing, while the majority of these drugs have been evaluated in preclinical studies and are not available in clinical practice. Currently available anti-myeloma agents, such as PIs, IMiDs, and anti-CD38 MoAb, work for the BM microenvironment and show an antimyeloma effect. ASCT is expected to not only have an antitumor effect through high-dose chemotherapy, but also help in the improvement of the BM microenvironment including immune reconstitution. There is no specific treatment for the inhibition of clonal evolution that shows acquired treatment resistance as yet, although treatment strategy, guided by the myeloma phenotype using flowcytometry, can be selected. In addition, maintenance of a deep response should be essential in preventing clonal evolution. The initial treatment strategies containing PIs, IMiDs, MoAbs, and ASCT should be necessary to overcome de novo drug resistance and prevent acquired drug resistance. It may be possible to select personalized treatment strategies through the analysis of clonal variation in each case using next generation sequencing. In the future, it is important not only to treat the myeloma cells themselves but also to develop a treatment strategy that considers the microenvironment and clonal evolution.

## Figures and Tables

**Figure 1 cancers-13-00215-f001:**
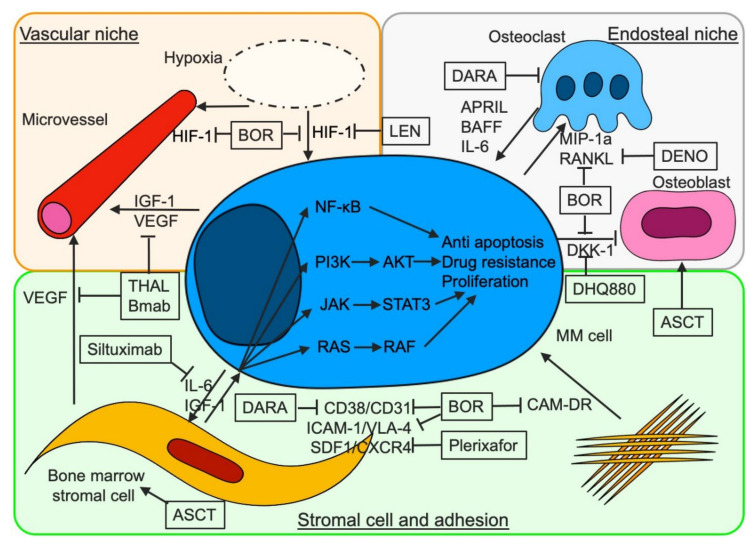
Therapeutics for bone marrow microenvironment associated with the progression and survival of myeloma cells. Stromal cells contribute to the proliferation, survival, and resistance against therapeutics in myeloma cells through the activation of several signaling pathways, cell adhesion-mediated drug resistance (CAM-DR), and soluble factor-mediated drug resistance (SFM-DR). BOR, DARA, and plerixafor overcome CAM-DR by inhibiting VLA-4, CD38, and CXCR4, respectively. BOR inhibits CD38-CD31-mediated adhesion of myeloma cells to bone marrow stromal cells (BMSCs) and increases CD38 expression on myeloma cells by decreasing BMSC CD31 expression. Under hypoxic conditions, HIF-1 promotes VEGF transcription, which induces the growth of microvessels and contributes to the proliferation of myeloma cells. IGF-1 contributes to the growth of microvessels through the activation of HIF-1alpha. Hypoxia is associated with stemness of myeloma cells. BOR suppresses transcription of HIF-1 alfa and indirectly inhibits myeloma angiogenesis via decreasing levels of VEGF. THAL inhibits angiogenesis by suppressing transcription of the *VEGF* gene. Bevacizumab suppresses the interaction between VEGF and VEGFR. Myeloma cells activate osteoclasts and inhibit osteoblasts, which constitute the endosteal niche. In addition, several cytokines from osteoclasts contribute to the proliferation of myeloma cells. BOR suppresses osteogenesis via inhibition of RANKL and DKK-1. DENO and BHQ088 inhibit RANKL and DDK-1, respectively. ASCT contributes to the improvement of BM environment by supplying mesenchymal cells and remodeling the endosteal niche. Mesenchymal stem cells, which are provided from autografts, contribute to the remodeling of bone marrow stromal cells and the activation of osteoblasts. BOR, bortezomib; THAL, thalidomide; IMiDs, immunomodulatory drugs; LEN, lenalidomide; DARA, daratumumab; DENO, denosumab; ASCT, autologous stem cell transplantation; Bmab, bevacizumab; CAM-DR, cell-adhesion mediated drug resistance; VEGF, vascular endothelial growth factor; IGF-1, insulin-like growth factor-1; HIF-1, hypoxia inducible factor-1; VLA-4, very late antigen 4; ICAM-1, intercellular adhesion molecule-1; CXCR4, C-X-C chemokine receptor type 4; SDF-1, stromal cell-derived factor-1; IL, interleukin; APRIL, a proliferation-inducing ligand; BAFF, B cell activating factor; RANKL, receptor activator of nuclear factor kappa-B ligand; MIP-1alpha, macrophage inflammatory protein 1alpha; DKK-1, Dickkopf-1.

**Figure 2 cancers-13-00215-f002:**
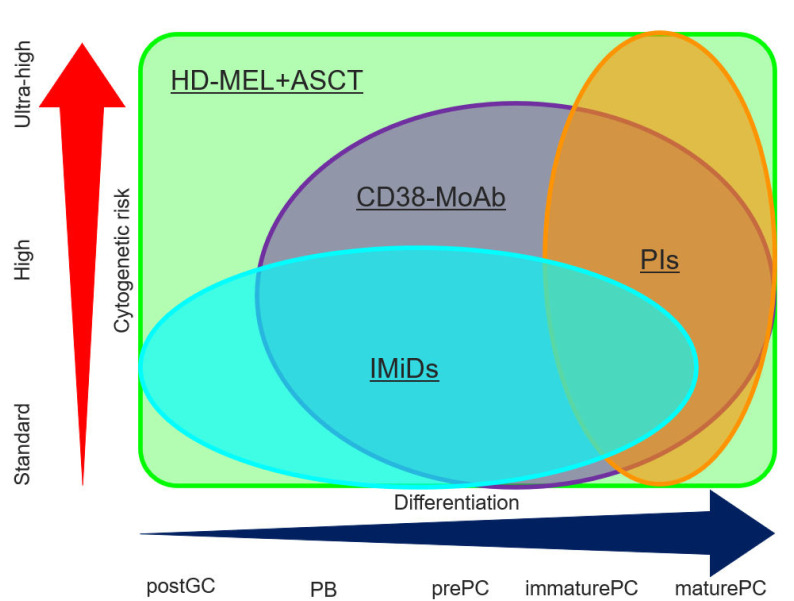
Activity of PIs, IMiDs, CD38 MoAB, and ASCT in terms of the differentiation of MM cells and cytogenetic risk. PIs contribute to anti-myeloma activity in more mature MM cells relatively independent from cytogenetic risk. IMiDs attack immature MM cells, but this is not enough to treat high-risk cytogenetic abnormality. CD38 MoAb can be used for CD38-positive MM cells excluding post GC cells and improves clinical outcome in the patients with HRCA through long-interval administration. A high dose of melphalan followed by ASCT kills MM cells independent of the differentiation of MM cells and cytogenetic risk. PIs, proteasome inhibitors; IMiDs, immunomodulatory drugs; MoAbs, monoclonal antibody agents; HD-MEL + ASCT, high dose melphalan followed by autologous stem cell transplantation; postGC, post germinal center; PB, plasmablast; PC, plasma cell.

**Figure 3 cancers-13-00215-f003:**
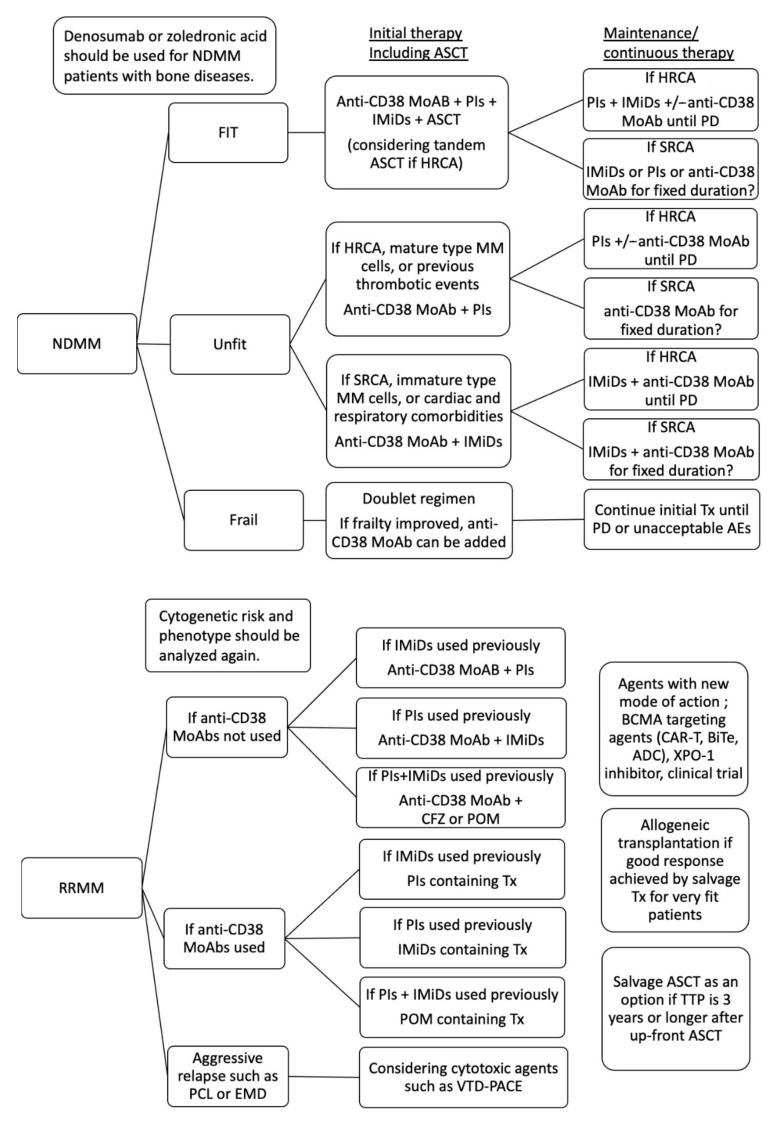
Treatment strategies for fit and unfit patients with NDMM, and RRMM, terminal phase MM. Multi-drug treatment including PIs, IMiDs, CD38 MoAb, and ASCT is essential for fit patients with NDMM. For unfit patients, who are eligible for intensive treatment including ASCT, CD38 MoAb plus PIs or IMiDs should be selected according to the differentiation of MM cells, cytogenetic risk and comorbidities. Maintenance or continuous therapy using two or three class agents until PD might be essential for patients with HRCA. Maintenance or continuous therapy using one class agents for fixed duration might be possible for patients with SRCA if MRD-negativity was achieved. For frail patients, doublet regimes might be selected considering general condition, comorbidity, and social background rather than cytogenetic risk. If frailty improved via the efficacy of chemotherapy, anti-CD38 MoAb can be added. Initial treatment should be continued until PD or unacceptable adverse events. When the disease is relapsed or refractory, cytogenetics and phenotype should be analyzed again. Anti-CD38 MoAbs should be used as second line of therapy if anti-CD38 MoAbs has not been administered yet. A high dose of carfilzomib or pomalidomide containing therapy might be active for patients with refractoriness for PIs and IMiDs. PIs containing therapy are selected as a second line of therapy if PIs have not been administered as initial therapy yet. IMiDs containing therapy are selected as a second line of therapy if IMiDs have not been administered as initial therapy yet. Salvage ASCT is considered when the time to progression from up-front ASCT is 3 years or more. Allogeneic transplantation is considered for very fit patients who achieved a good response using salvage chemotherapy. Cytotoxic agents-containing therapy, such as VTD-PACE, might be active for aggressive relapse, such as extramedullary disease and plasma cell leukemia. Agents with a new mode of action, BCMA-targeting therapy, XPO-1 inhibitors, and clinical trials are considered when they are available. NDMM, newly diagnose multiple myeloma; RRMM relapse or refractory multiple myeloma, HRCA, high risk cytogenetic abnormality; SRCA, standard risk cytogenetic abnormality; PD, progressive disease; PIs, proteasome inhibitors; IMiDs, immunomodulatory drugs; MoAbs, monoclonal antibody agents; ASCT, autologous stem cell transplantation; CFZ, carfilzomib; POM, pomalidomide; EMD, extramedullary disease; PCL, plasma cell leukemia; TTNT, time to progression; XPO-1, exportin-1; BCMA, B cell mature antigen, CAR-T, chimeric antigen receptor-T cell; ADC, antibody-drug conjugate; bispecific T-cell engager; Tx, treatment.

**Table 1 cancers-13-00215-t001:** Clinical trials of new agents for target concerning bone marrow microenvironments in multiple myeloma (MM). BOR, bortezomib; THAL, thalidomide; DEX, dexamethasone; PCB, placebo; CXCR4, C-X-C chemokine receptor type 4; IL-6, interleukin-6; IGF-1R, insulin-like growth factor-1 receptor; VEGF, vascular endothelial growth factor; VEGFR, vascular endothelial growth factor receptor; RANKL, receptor activator of nuclear factor kappa-B ligand; DKK-1, Dickkopf-1; BAFF, B cell activating factor; NDMM, newly diagnosed multiple myeloma; RRMM, relapsed or refractory multiple myeloma; ORR, overall response rate; CBR, clinical benefit rate; PFS, progression free survival; EFS, event free survival; mo, months, NS = not significant.

Target	Experimental Arm	Control Arm	Disease Status	Phase	Endopoint	Outcome	References
Stromal cell and adhesion							
CXCR4	Plerixafor + BOR	–	RRMM prior BOR	2	ORR	45% (CBR = 60.6%)	[14]
IL-6	Siltuximab + BOR	–	RRMM	2	ORR	0% (siltuximab only), 8% (plus DEX)	[15]
IL-6	Siltuximab ± DEX	–	RRMM	2	PFS	8.0 vs. 7.6 mo (*p* = 0.345)	[16]
IGF-1R	Figitumumab ± DEX	–	RRMM	1	ORR	33%	[17]
Vascular niche							
VEGF-A	Bevacizumab + THAL	–	RRMM	2	ORR	33%, EFS = 37−369 days	[18]
VEGF-A	Bevacizumab + BOR	BOR	RRMM	2	PFS	6.2 vs. 5.1 mo (*p* = 0.28)	[19]
VEGFR	Sorafenib	–	RRMM	2	ORR	9% (CBR = 18%)	[20]
VEGFR	Sorafenib	–	RRMM	2	ORR	0%	[21]
VEGFR-2	Vandetanib	–	RRMM	2	ORR	0%	[22]
Endosteal niche							
RANKL	Denosumab	Zoledronic acid	NDMM	3	Time to skeletal events, PFS	22.8 vs. 24.0% (*p* = 0.01, non-inferior), PFS = 46.1 vs. 35.4 mo (*p* = 0.036)	[23]
DKK-1	BHQ880	–	RRMM	1b	ORR	15%, CBR = 23% (10 mg/kg)	[24]
BAFF	Tabalumab + BOR + DEX	BOR + DEX	RRMM	2	PFS	6.6 (100 mg) vs. 7.5 (300 mg) vs. 7.6 mo (PCB) (*p* = NS)	[25]

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
