# Peer review of "Treatment Strategies Considering Micro-Environment and Clonal Evolution in Multiple Myeloma"

_cancers, 2021, doi:10.3390/cancers13020215_

Round 1

Reviewer 1 Report

This will be of interest to field for a review of treatment strategies as it relates to both de novo and acquired drug resistance.   The focus on clonal evolution and acquired resistance was well organized  

Comments line 48-50: Review refers to overlapping phenotypes in the vascular and endosteal niche of the MM cells but states that the vascular niche supports stemness more detail of the hypoxic and vascular niche as it relates to the MM stemcell population may add clarity to this statement

Line 58 Target molecules or perhaps target cell surface receptors is a better description.  Molecules typically refers to chemical compound.  Confused with description of non-genetic diversity as a mechanism underlying acquired drug resistance for moABs. Is loss of cell surface expression a change in genotype?  I think needs to be more clarity or details presented here.

Line 138 IGF1 is considered a therapeutic molecule do authors mean therapeutic target?

When talking about O2 content of bone marrow some numbers of O2 content in the niche (vascular versus endosteal) would be helpful

Manuscript needs to be carefully edited,

Author Response

Reviewer 1

Comments and Suggestions for Authors

This will be of interest to field for a review of treatment strategies as it relates to both de novo and acquired drug resistance.   The focus on clonal evolution and acquired resistance was well organized  

Comments line 48-50: Review refers to overlapping phenotypes in the vascular and endosteal niche of the MM cells but states that the vascular niche supports stemness more detail of the hypoxic and vascular niche as it relates to the MM stem cell population may add clarity to this statement

Response: Thank you for your suggestion. In the revised manuscript we have added the following details to the “vascular niche” and “endosteal niche” as below:

Line 189-190.

MM associated niches are categorized into “vascular niche” and “endosteal niche”.

->

MM associated niches are categorized into “vascular niche” and “endosteal niche” which are associated vasculature and osteopoiesis, respectively.

Line 58 Target molecules or perhaps target cell surface receptors is a better description.  Molecules typically refers to chemical compound.  Confused with description of non-genetic diversity as a mechanism underlying acquired drug resistance for moABs. Is loss of cell surface expression a change in genotype?  I think needs to be more clarity or details presented here.

Response: Thank you for your suggestion. CD38 and SLAMF-7 are identified with not molecule but cell surfaced protein. In addition, loss of surface expression of CD38 and SLAMF-7 is not associated with genetic change but non-genetic diversity, which is described in “5.7Nongenetic clonal diversity" in detail. Therefore, we have changed from “target molecule” into “target” and added that the expression of CD38 and SLAMF-7 depends on nongenetic clonal diversity as bellow:

Line 213-216.

Hypoxia prevents myeloma cells from maturing and might reduce the expression of signaling lymphocytic activation molecule family member 7 (SLAMF7) and CD38, which are the target molecules of elotuzumab (ELO) and DARA or isatuximab (ISA), respectively.

->

Hypoxia prevents myeloma cells from maturing and might reduce the expression of signaling lymphocytic activation molecule family member 7 (SLAMF7) and CD38, which are the target of elotuzumab (ELO) and DARA or isatuximab (ISA), depending on non-genetic diversity which is described “5.7 Nongenetic clonal diversity” in detail.

Line 138 IGF1 is considered a therapeutic molecule do authors mean therapeutic target?

Response: Thank you for your suggestion. I consider that IGF-1R is a therapeutic target. In the revised manuscript we changed therapeutic target is not IGF-1 but IGF-1R as below:

Line 168-169.

Thus, IGF-1 is associated with angiogenesis and survival of myeloma cell, and is considered a therapeutic molecule.

 ->

Thus, interaction between IGF-1 and IGF-1R is associated with angiogenesis and survival of myeloma cell, and IGF-1R is considered a therapeutic target.

When talking about O2 content of bone marrow some numbers of O2 content in the niche (vascular versus endosteal) would be helpful

Response: Thank you for your suggestion. Harrison JS, et al. demonstrated that mean O2 saturation in bone marrow is 87.5% +/- 1.1% when those in peripheral blood is 99% in healthy volunteers. In the revised manuscript we added O2 saturation in bone marrow and peripheral blood as below:

Line 206-208.

Mean oxygen saturation in bone marrow is 87.5% +/- 1.1% when those in peripheral blood is 99% in healthy volunteers [70].

Reviewer 2 Report

Dr Suzuki and Collegues have provided a comprehensive review on a relevant topic related to the role of the bone marrow (BM) milieu in supporting multiple myeloma (MM) pathogenesis and clonal evolution in MM. Impotantly, the therapeutic strategies that have been identified based on the preclinical studies addressing the role of BM in MM have been emphasized. The review article is well presented and structured. References are appropriate. The Reader is also guided throughout the text by clear and well presented Figures. 

The present Reviewer would ask to add a few relevant references within the field. For instance:

- Cell Rep, 2014;9:118-128. doi: 10.1016/j.celrep.2014.08.042 (targeting SDF1 within the BM niche to halt MM cell dissemination and disease progression)

Author Response

Reviewer 2

Comments and Suggestions for Authors

Dr Suzuki and Collegues have provided a comprehensive review on a relevant topic related to the role of the bone marrow (BM) milieu in supporting multiple myeloma (MM) pathogenesis and clonal evolution in MM. Impotantly, the therapeutic strategies that have been identified based on the preclinical studies addressing the role of BM in MM have been emphasized. The review article is well presented and structured. References are appropriate. The Reader is also guided throughout the text by clear and well presented Figures. 

The present Reviewer would ask to add a few relevant references within the field. For instance:

- Cell Rep, 2014;9:118-128. doi: 10.1016/j.celrep.2014.08.042 (targeting SDF1 within the BM niche to halt MM cell dissemination and disease progression)

Response: Thank you for your suggestion. In the revised manuscript we added reference depending on your recommendation in “3.2 Hypoxia in myeloma” as below:

Line213-216.

In addition, hypoxia increases C-X-C chemokine receptor type 4 (CXCR4) expression on MM cell and decreases SDF-1 from stromal cells, inducing MM cells to migrate to new BM niches [63, 64].

->

In addition, hypoxia increases C-X-C chemokine receptor type 4 (CXCR4) expression on MM cell and decreases SDF-1 from stromal cells, inducing MM cells to migrate to new BM niches [76-78].

Line 813-815.

  1. Roccaro, AM.; Sacco, A.; Purschke, WG.; Moschetta, M.; Buchner, K.; Maasch, C.; Zboralski, D.; Zöllner, S.; Vonhoff, S.; Mishima, Y.; Maiso, P.; et al. SDF-1 inhibition targets the bone marrow niche for cancer therapy. Cell Rep. 2014, 9, 118-128.

Reviewer 3 Report

Kazuhito Suzuki et al. and colleagues report the role of anti-myeloma agents for microenvironment and clonal evolution and treatment strategies to overcome drug resistance. The manuscript is of interest, However, major issues need to be addressed.

1.The rationale of why the authors came up with this review.

2. What is the information that is not exactly available that motivated the authors to come up with this information? What are the current caveats and how do the authors highlight the current research in answering them? If not they need to address in future directions.

3. I personally miss some important insights about the biological, dynamic and context-dependent role of the junctional adhesion molecules mediating the vicious cycle existing between the myeloma cells and the neighbourhood. Indeed, this theragnostic potential of interfering with the adhesion system has been observed mostly in preclinical models, not in patients and, therefore, must be interpreted with caution. Nevertheless, our new findings may point towards a potential Achilles’ heel of multiple myeloma that might be exploited therapeutically in the future. 

4.I would consider restructuring the figures (i.e. fig 1 and 2 can be incorporated in one figure and an additional one (or a table) might suffice to describe the therapeutic windows: the underlying message here is that more precision and individualized approaches need to be tested in well-designed clinical trials – a challenge, but I would be interested in their perspective of how this might be done.

5. Our understanding of factors influencing prognosis in MM has advanced considerably. We now recognize the contribution of a range of features including patient’s baseline risk stratification, disease biology, genetic lesions, imaging findings, and depth of response. Therefore, it is reasonable to design tailored clinical trials aimed to stratify patients differentially according to disease risk. Remarkable efforts have been attempted in order to translate these unmet clinical needs to bedside-approaches (PMID: 31323969). Can the author comment on this issue and slightly integrate the therapeutic algorithm and envisioned strategy in light of the pragmatic Integrated approach to MM patients according to the clinical risk profile that can be envisioned? In the mentioned available data (PMID: 31323969), a clinical trial design proposal for a risk-driven personalized approach might also foster a snapshot on therapeutic targets within the tumor milieu in multiple myeloma.

Author Response

Reviewer 3

Comments and Suggestions for Authors

Kazuhito Suzuki et al. and colleagues report the role of anti-myeloma agents for microenvironment and clonal evolution and treatment strategies to overcome drug resistance. The manuscript is of interest, However, major issues need to be addressed.

  1. The rationale of why the authors came up with this review.

Response: Thank you for your suggestion. Multiple myeloma is generally uncurable hematological malignancy due to de novo and acquired drug resistance although the prognosis of myeloma patients is getting better using proteasome inhibitor, immunomodulatory drugs, and monoclonal antibody in two last decade. Overcoming de novo and acquired drug resistance plays an important role to lead clinical cure in myeloma patients. Therefore, I described this review to summarize mechanism of drug resistance and treatment strategy to overcome drug resistance. In the revised manuscript we added in “Introduction” as below:

Line73-76.

Finally, four classes of anti-myeloma agents are available: proteasome inhibitors, immune modulatory drugs, monoclonal antibodies, and cytotoxic agents.

->

Finally, four classes of anti-myeloma agents are available: proteasome inhibitors, immune modulatory drugs, monoclonal antibodies, and cytotoxic agents, but MM is an uncurable hematological malignancy due to de novo and acquired drug resistance although the prognosis of myeloma patients is getting better using these anti-myeloma agents. Overcoming de novo and acquired drug resistance plays an important role to lead clinical cure in myeloma patients.

  1. What is the information that is not exactly available that motivated the authors to come up with this information? What are the current caveats and how do the authors highlight the current research in answering them? If not they need to address in future directions.

Response: Thank you for your suggestion. Several new drugs are developed to overcome de novo drug resistance, such as proteasome inhibitors and EZH2 inhibitors to overcome CAM-DR, and anti-IL-6 monoclonal antibodies and IGF-1R inhibitors to overcome SFM-DR, in preclinical study although several of those has not been shown efficacy in clinical trial and available in clinical practice. Meanwhile, treatment to prevent and overcome acquired drug resistance has already been one of unmet medical needs. In future, treatment strategy to prevent clonal evolution and overcome acquired drug resistance including terminal phase of myeloma will be necessary. In the revised manuscript we added a new section “6.5 Future direction” in “6. Overall treatment strategy considering microenvironment and clonal evolution” as below:    

Line548-558

In future, treatment with new mode of action, such as chimeric antigen receptor-T cell therapy (CAR-T) [193], antibody-drug conjugate (ADC) [194], bispecific T-cell engager (BiTE) for B cell mature antigen (BCMA) [195], exportin-1 (XPO-1) inhibitor [196] and melflufen [197] may be active for the patients with terminal phase of MM.

->

6.5  Future direction

Several new drugs are developed to overcome de novo drug resistance, such as proteasome inhibitors and EZH2 inhibitors to overcome CAM-DR, and anti-IL-6 monoclonal antibodies and IGF-1R inhibitors to overcome SFM-DR, in preclinical study although several of those has not been shown efficacy in clinical trial. In addition, treatments to prevent and overcome acquired drug resistance has already been one of unmet medical needs as well. In future, treatment strategy to prevent clonal evolution and overcome acquired drug resistance including terminal phase of myeloma will be necessary. Treatment with new mode of action, such as chimeric antigen receptor-T cell therapy (CAR-T) [193], antibody-drug conjugate (ADC) [194], bispecific T-cell engager (BiTE) for B cell mature antigen (BCMA) [195], exportin-1 (XPO-1) inhibitor [196] and melflufen [197] may be active for the patients with terminal phase of MM.

  1. I personally miss some important insights about the biological, dynamic and context-dependent role of the junctional adhesion molecules mediating the vicious cycle existing between the myeloma cells and the neighbourhood. Indeed, this theragnostic potential of interfering with the adhesion system has been observed mostly in preclinical models, not in patients and, therefore, must be interpreted with caution. Nevertheless, our new findings may point towards a potential Achilles’ heel of multiple myeloma that might be exploited therapeutically in the future. 

Response: Thank you for your suggestion. I agree with your opinion concerning treatment for adhesion molecules in MM. Several new drugs are developed to overcome de novo drug resistance, such as proteasome inhibitors and EZH2 inhibitors to overcome CAM-DR, and anti-IL-6 monoclonal antibodies and IGF-1R inhibitors to overcome SFM-DR, in preclinical study although several of those has not been shown efficacy in clinical trial and available in clinical practice. In future, I am looking forward to evaluating efficacy of several drugs for de novo drug resistance in clinical trial and practice. In the revised manuscript we added a new section “6.5 Future direction” in “6. Overall treatment strategy considering microenvironment and clonal evolution” as below (same as response statement for question 2. as above):

Line548-558.

In future, treatment with new mode of action, such as chimeric antigen receptor-T cell therapy (CAR-T) [193], antibody-drug conjugate (ADC) [194], bispecific T-cell engager (BiTE) for B cell mature antigen (BCMA) [195], exportin-1 (XPO-1) inhibitor [196] and melflufen [197] may be active for the patients with terminal phase of MM.

->

6.5  Future direction

Several new drugs are developed to overcome de novo drug resistance, such as proteasome inhibitors and EZH2 inhibitors to overcome CAM-DR, and anti-IL-6 monoclonal antibodies and IGF-1R inhibitors to overcome SFM-DR, in preclinical study although several of those has not been shown efficacy in clinical trial. In addition, treatments to prevent and overcome acquired drug resistance has already been one of unmet medical needs as well. In future, treatment strategy to prevent clonal evolution and overcome acquired drug resistance including terminal phase of myeloma will be necessary. Treatment with new mode of action, such as chimeric antigen receptor-T cell therapy (CAR-T) [193], antibody-drug conjugate (ADC) [194], bispecific T-cell engager (BiTE) for B cell mature antigen (BCMA) [195], exportin-1 (XPO-1) inhibitor [196] and melflufen [197] may be active for the patients with terminal phase of MM.

4.I would consider restructuring the figures (i.e. fig 1 and 2 can be incorporated in one figure and an additional one (or a table) might suffice to describe the therapeutic windows: the underlying message here is that more precision and individualized approaches need to be tested in well-designed clinical trials – a challenge, but I would be interested in their perspective of how this might be done.

Response: Thank you for your suggestion. I combined figure 1 and 2 into new figure 1. Figure 1 show that Therapeutics for bone marrow microenvironment associated with progression and survival of myeloma cells. I added table 1 which showed the clinical trial of therapeutics, excluding proteasome inhibitors, IMiDs, and anti-CD38 monoclonal antibodies, for bone marrow microenvironment. In the revised manuscript, we changed figure legends 1 as below:

Line83-120.

The relationship between MM cells and BMSC, vascular niche, and endosteal niche are summarized in Figure 1. Treatments for microenvironments in MM are shown in Figure 2.

->

The relationship between MM cells and BMSC, vascular niche, and endosteal niche and treatment for microenvironment are summarized in Figure 1. Clinical trial of new agents for target concerning bone marrow microenvironments in MM are shown in table 1.

  1. Our understanding of factors influencing prognosis in MM has advanced considerably. We now recognize the contribution of a range of features including patient’s baseline risk stratification, disease biology, genetic lesions, imaging findings, and depth of response. Therefore, it is reasonable to design tailored clinical trials aimed to stratify patients differentially according to disease risk. Remarkable efforts have been attempted in order to translate these unmet clinical needs to bedside-approaches (PMID: 31323969). Can the author comment on this issue and slightly integrate the therapeutic algorithm and envisioned strategy in light of the pragmatic Integrated approach to MM patients according to the clinical risk profile that can be envisioned? In the mentioned available data (PMID: 31323969), a clinical trial design proposal for a risk-driven personalized approach might also foster a snapshot on therapeutic targets within the tumor milieu in multiple myeloma.

Response: Thank you for your suggestion. I changed from figure 4 into new figure 3 about treatment strategy. In the revised manuscript, we changed figure 3 as below:

Line 423-463.

Association between the activity of treatments, including PIs, IMiDs, anti CD38 MoAb, and ASCT, cytogenetic risk and maturation of MM cells are summarized in figure 3. Treatment strategies for fit and unfit patients with NDMM are shown in figure 4.

->

Association between the activity of treatments, including PIs, IMiDs, anti CD38 MoAb, and ASCT, cytogenetic risk and maturation of MM cells are summarized in figure 2 [128, 172, 173]. Treatment strategies for fit and unfit patients with NDMM, and RRMM, terminal phase MM are shown in figure 3.

Round 2

Reviewer 1 Report

The authors have clarified my concerns

Author Response

Thank you for the comment.

Reviewer 3 Report

The authors have clarified several of the questions I raised in my previous review. Most of the major problems have been addressed by this revision. As I stated in my previous review, I deem it likely that all the suggested point and issues, that have been collectively solved by the added paragraphs increase the impact of the manuscript for the scientific community. Moreover, I feel fair to mention the reference PMID: 31323969, given the content of the added algorithm approach proposed in this revised version by the authors, that largely deal with a pragmatic integrated approach to MM patients according to the clinical risk profile, at least partially inspired by the data presented in PMID: 31323969. The revision process is for sure also subjective and the final decision is for the Editorial office.

Author Response

Thank you for the comment. I consider that this treatment strategy is best as best my knowledge. Therefore, I did not change treatment algorithm.